# Metabolic Stability and Metabolite Identification of N-Ethyl Pentedrone Using Rat, Mouse and Human Liver Microsomes

**DOI:** 10.3390/pharmaceutics16020257

**Published:** 2024-02-09

**Authors:** Alexandre Barcia Godoi, Natalícia de Jesus Antunes, Kelly Francisco Cunha, Aline Franco Martins, Marilyn A. Huestis, Jose Luiz Costa

**Affiliations:** 1Centro de Informação e Assistência Toxicológica (CIATox) de Campinas, Universidade Estadual de Campinas (UNICAMP), Campinas 13083-859, SP, Brazil; bdgalexandre@gmail.com (A.B.G.); k180042@dac.unicamp.br (K.F.C.); alinefmartins14@gmail.com (A.F.M.); josejlc@unicamp.br (J.L.C.); 2Faculdade de Ciências Médicas, Universidade Estadual de Campinas (UNICAMP), Campinas 13083-859, SP, Brazil; 3Institute of Emerging Health Professions, Thomas Jefferson University, Philadelphia, PA 19107, USA; marilyn.huestis@gmail.com; 4Faculdade de Ciências Farmacêuticas, Universidade Estadual de Campinas (UNICAMP), Campinas 13083-859, SP, Brazil

**Keywords:** new psychoactive substances, N-ethyl pentedrone, NEP, metabolic stability, metabolites, liver microsomes, in vitro evaluation

## Abstract

New Psychoactive Substances (NPSs) are defined as a group of substances produced from molecular modifications of traditional drugs. These molecules represent a public health problem since information about their metabolites and toxicity is poorly understood. N-ethyl pentedrone (NEP) is an NPS that was identified in the illicit market for the first time in the mid-2010s, with four intoxication cases later described in the literature. This study aims to evaluate the metabolic stability of NEP as well as to identify its metabolites using three liver microsomes models. To investigate metabolic stability, NEP was incubated with rat (RLM), mouse (MLM) and human (HLM) liver microsomes and its concentration over time evaluated by liquid chromatography–mass spectrometry. For metabolite identification, the same procedure was employed, but the samples were analyzed by liquid chromatography–high resolution mass spectrometry. Different metabolism profiles were observed depending on the model employed and kinetic parameters were determined. The in vitro NEP elimination half-lives (t_1/2_) were 12.1, 187 and 770 min for the rat, mouse and human models, respectively. Additionally, in vitro intrinsic clearances (Cl _int, in vitro_) were 229 for rat, 14.8 for mouse, and 3.6 μL/min/mg in the human model, and in vivo intrinsic clearances (Cl _int, in vivo_) 128, 58.3, and 3.7 mL/min/kg, respectively. The HLM model had the lowest rate of metabolism when compared to RLM and MLM. Also, twelve NEP metabolites were identified from all models, but at different rates of production.

## 1. Introduction

The emergence of New Psychoactive Substances (NPSs) has challenged and alarmed health care systems, toxicologists and drug control authorities [1]. These substances, synthetic drugs produced by structural modifications of well-known drugs of abuse, are produced with the aim of evading the legislation of chemical substance control [2]. Thus, the new structures are a toxicological challenge, since their physicochemical and pharmacological properties are altered, potentially leading to severe cases of intoxication [3].

Due to the large number of NPSs described in recent years, these substances are organized into classes based on their molecular structures [4]. Synthetic cathinones are among the most prevalent NPS worldwide, with a high toxicity potential. Synthetic cathinones can significantly increase excitatory synaptic activity, mainly through dopamine, serotonin and noradrenaline neurotransmitters [5]. N-ethyl pentedrone (NEP) is a synthetic cathinone first identified in the mid-2010s, with reports about its toxic effects in subsequent years. Recently, four NEP intoxication cases have been published, all with renal and heart failure, diffuse alveolar hemorrhage and death [6,7,8,9]. These publications stated that NEP is commonly ingested by inhalation, but there is no information on ingested doses. The website The Drug Classroom describes user reports of 20–50 mg NEP doses [10]. The Center for Forensic Science Research and Education (CFSRE) reported NEP seizures by police and its presence in biological materials in the United States [11], and the substance was detected by police in the Northeast region of Brazil [12]. Despite knowledge of NEP for almost 10 years and case reports of important neurological, renal, cardiac and pulmonary toxic effects, limited toxicological studies have been conducted.

In vitro studies can establish important pharmacokinetic parameters such as elimination half-life (t_1/2_) and intrinsic clearance (Cl _int, in vivo_), and predict in vivo kinetic behaviors through enzymes contained in the liver microsomes, including P450 cytochrome (CYP450), carboxyl esterases and UDP glucuronyltransferases (UGT) [13]. Furthermore, identifying phase I and II metabolites and their formation rates following the drug’s biotransformation also provides essential information for understanding its toxicity [14]. However, in vitro studies with interspecies comparison are crucial to drug characterization, allowing a suitable evaluation of the best animal model for complementary toxicokinetic and toxicodynamic studies [15].

This study aims to demonstrate, for the first time, in vitro interspecies NEP metabolism by rat (RLM), mouse (MLM) and human (HLM) liver microsomes.

## 2. Materials and Methods

### 2.1. Chemical and Reagents

Methanol, acetonitrile, ammonium formate, glucose-6-phosphate, β-nicotinamide adenine dinucleotide phosphate hydrate (NADP^+^), uridine 5′-diphosphoglucuronic acid trisodium salt (UDPGA), adenosine 3′-phosphate 5′-phosphosulfate triethylammonium salt (PAPS), S-(5′-adenosyl)-L-methionine *p*-toluenesulfonate salt (SAM), magnesium chloride hexahydrate and sodium citrate tribasic dihydrate were purchased from Sigma-Aldrich (St. Louis, MI, USA). Formic acid was acquired from Scharlab (Sentmenat, Barcelona, Spain) and Gentest™ 0.5 M phosphate buffer pH 7.4 and glucose-6-phosphate dehydrogenase from Corning (Woburn, MA, USA). Ultrapure water was obtained using a Mili-Q RG system from the company Milipore (Burlington, MA, USA). NEP hydrochloride reference material (98% purity) was purchased from Cayman Chemical (Ann Arbor, MI, USA) and a stock solution prepared at 1 mg/mL (4.1 mM free base) in methanol. NEP working solutions of 40 and 732 μM were also prepared by appropriate dilution of NEP stock solution in methanol. Dapaconazole, the positive control for the experiment, was supplied by Biolab Farmacêutica Ltda. (São Paulo, Brazil) and a stock solution prepared at 1 mg/mL (2.4 mM) in methanol. A dapaconazole working solution of 40 μM was prepared from a dapaconazol stock solution in methanol.

### 2.2. Microsome Preparation and Incubation

Pooled HLM containing 20 mg/mL microsomal proteins and 270 pmol CYP450/mg protein was obtained from Sigma-Aldrich. RLM and MLM liver microsomes were prepared from 5-pooled animal liver by differential ultracentrifugation [16,17]. Healthy animals were sacrificed to perform other experiments and their livers were removed and placed in 0.05 mol/L Tris-HCl buffer (pH 7.4), containing 0.15 mol/L KCl. The livers were minced with scissors and washed three times with the buffer. The slices added with 20 mL of the buffer were ground in a Potter-type homogenizer in three cycles, each cycle comprising 3 grindings of 1 min at 1000 rpm. The homogenate was centrifuged at 10,000× *g* for 15 min at 4 °C. The supernatant was ultracentrifuged at 100,000× *g* for 60 min at 4 °C to obtain the microsomal pellet, which was then resuspended in HEPES-HCl buffer (pH 7.4; 0.05 mol/L) containing 20% glycerol and 0.001 mol/L EDTA and stored at −80 °C until use. Protein concentration was determined by the Bradford method [18,19].

For the determination of metabolic stability, incubations followed the good practices guideline for metabolism studies [20]. Ten microliters of 40 µM NEP was added to 1.5 mL propylene tubes and dried under nitrogen. Then, 100 μL of a NADPH-regenerating system, containing 1.1 mM NADP^+^, 10 mM glucose-6-phosphate, 1 U/mL glucose-6-phosphate dehydrogenase, 5 mM sodium citrate and 66 mM magnesium chloride in 100 mM phosphate buffer, pH 7.4 was added to the tubes. The solutions were then pre-incubated for 5 min in a MTC 100 thermo shaker incubator (Miulab, Hangzhou, ZJ China) at 300 rpm and 37 °C. To start the reactions, aliquots of 100 μL of RLM, MLM and HLM at 1 mg/mL of protein were added to the pre-incubated shaking tubes, yielding NEP and protein final concentrations of 2 μM and 0.5 mg/mL, respectively. After 0, 3, 5, 15, 30 and 60 min, the reactions were stopped by adding 400 μL ice-cold acetonitrile to the medium. Then, the samples were vortexed for 5 min in a BenchMixer™ XL (Benchmark, NJ, USA) and centrifuged at 12,000× *g* for 15 min at 4 °C (Hettich^®^ Universal 320 R, Tuttlingen, BW, Germany). The supernatants were diluted 1:9 (*v*/*v*) in ultrapure water and 200 μL was transferred to a 96-well plate. Aliquots (2 μL) were injected into a liquid chromatograph in tandem with a mass spectrometer (LC-MS/MS). Positive controls were prepared by incubating dapaconazole under the same condition as NEP (incubation concentration 2 μM). Negative controls were prepared by incubating NEP in buffer solution, in the absence of microsome and cofactor solutions. The metabolism rate was measured by analyzing the decreasing areas of the chromatographic peaks of the analytes at different incubation times.

For phase I metabolite elucidation, incubations were achieved in just one replicate, employing the same method described above. Ten microliters of 732 μM NEP was added to 1.5 mL propylene tubes and dried under nitrogen. Then, 100 μL of a NADPH-regenerating system, containing 1.1 mM NADP^+^, 10 mM glucose-6-phosphate, 1 U/mL glucose-6-phosphate dehydrogenase, 5 mM sodium citrate and 66 mM magnesium chloride in 100 mM phosphate buffer, pH 7.4, was added to the tubes. The tubes were pre-incubated for 5 min in an MTC 100 thermo shaker incubator at 300 rpm and 37 °C. In order to start the reactions, aliquots of 100 μL of RLM, MLM and HLM at 5 mg/mL were added into the pre-incubated shaking tubes, achieving NEP and protein final concentrations of 37 μM and 2.5 mg/mL, respectively. The metabolism reactions were stopped after 0, 15, 30, and 60 min by adding 400 μL ice-cold acetonitrile to the medium. The samples were mixed for 5 min and centrifuged at 12,000× *g* for 15 min at 4 °C. Supernatants were transferred to vials (200 μL) and 4 μL was injected into a liquid chromatograph in tandem with a high-resolution mass spectrometer (LC-HRMS).

For phase II metabolite elucidation, incubations were prepared following the conditions mentioned previously. Ten microliters of 732 μM NEP was added to 1.5 mL propylene tubes and dried under nitrogen. Then, 100 μL of 13.70 mM UDPGA, 0.49 mM PAPS, and 3.79 mM SAM in 100 mM phosphate buffer, pH 7.4, were added to the tubes in the presence (phase I followed by phase II) and absence (only phase II) of phase I cofactors, in order to access the phase II dependency of phase I metabolism. The tubes were pre-incubated for 5 min in an MTC 100 thermo shaker incubator at 300 rpm and 37 °C. In order to start the reactions, aliquots of 100 μL of RLM, MLM and HLM at 5 mg/mL of protein were added in the pre-incubated shaking tubes, achieving an NEP and protein final concentration of 37 μM and 2.5 mg/mL, respectively. The metabolism reactions were stopped after 0, 15, 30, 60 min by adding 400 μL ice-cold acetonitrile to the medium. The samples were mixed for 5 min and centrifuged at 12,000× *g* for 15 min at 4 °C. The supernatants were transferred to vials (200 μL) and 4 μL was injected into an LC-HRMS.

### 2.3. Detection of NEP by LC-MS/MS

Detection of NEP was performed in a Nexera HPLC chromatographic system coupled to a LCMS8045 triple quadrupole mass spectrometer (Shimadzu, Kyoto, Japan). Chromatographic separation was performed with a Raptor™ Biphenyl (2.1 × 100 mm, 2.7 μm) at 40 °C. The run was conducted with a flow rate of 0.4 mL/min, employing mobile phases constituted by water (A) and methanol (B) both added to 2 mM ammonium formate and 0.1% (*v*/*v*) formic acid. The applied gradient program started at 20% B up to 0.2 min, ramping to 95% B up to 2 min, maintaining this proportion for 1 min, and finally returning to the initial condition of 20% B up to 3.2 min, keeping this proportion until 5 min. The ionization source employed was electrospray (ESI) operating in positive mode. The analysis was performed in multiple reaction monitoring (MRM) mode. For NEP, a quantifying transition (206.1 > 130.0 *m*/*z*, using a collision energy (CE) of 32 eV) and a second confirmatory transition (206.1 > 188.0 *m*/*z*, using a CE of 15 eV) were monitored. For dapaconazole, only one transition was monitored (415.0 > 159.0 *m*/*z*, using a CE of 32 eV). Data acquisition was executed using the LabSolutions software version 5.114 (Shimadzu, Kyoto, Japan).

### 2.4. Identification of NEP and Its Metabolites by LC-HRMS

Determination of NEP and its metabolites was performed on a Nexera HPLC chromatographic system coupled to a LCMS9030 quadrupole-time-of-flight (QToF) analyzer (Shimadzu, Kyoto, Japan) with a ESI ionization source, operating in positive mode. Separation was executed in a gradient program using a Cortecs T3 C18 column (2.1 × 150 mm, 2.7 μm) at 40 °C. Mobile phases consisted of water (A) and methanol (B) added to 0.1% formic acid with a flow rate of 0.3 mL/min. Chromatography began with 20% B for 1 min, ramping to 95% B by 18 min, holding for 3 min and finally returning to initial conditions by 21.1 min, and maintained until 26 min. Spectra were obtained in data-dependent acquisition (DDA) mode with a full-scan MS mass range from 80 to 400 *m*/*z* and for the dependent events (MS/MS) from 60 to 300 *m*/*z* with a CE in spread mode of 25 ± 15 eV. Before all data acquisition, the mass spectrometer was calibrated to guarantee mass resolution and accuracy. For such, sodium iodide (Na-(NaI)_5_) was employed as a reference standard, monitoring *m*/*z* of 1971.614356, considering a maximum mass error of 1 ppm.

### 2.5. Data Analysis

Kinetic enzymatic determination of NEP was calculated using version 9.0 GraphPad Prism software (GraphPad Software, San Diego, CA, USA). Through the decay curves along the times of incubation in the different microsomal medium tested, it was possible to deduct t_1/2_ using the following equation [21]:$$t_{1/2}=\frac{\ln2}{\mathrm{slope}(k)}$$
where k is the slope of the log-linear regression graph of the drug remaining percentage as a function of time.

Furthermore, it was possible to calculate Cl _int, in vitro_ through the following equation [22]:$${Cl}_{int,\;in\;vitro}=\frac{0.693}{t_{1/2}}\times \frac{V_{incubation}}{m_{microsomes}}$$
where V_incubation_ is the volume of the incubation medium in μL and m_microsomes_ is the mass of microsomal proteins added to the incubation solution in mg.

Applying the determined Cl _int, in vitro_, it was also possible to calculate the in vivo intrinsic clearance (Cl _int, in vivo_) as follows:$${Cl}_{int,\;in\;vivo}={Cl}_{int,\;in\;vitro}\times \frac{m_{microsomes}}{g_{liver}}\times \frac{m_{liver}}{{kg}_{per\;body\;weight}}$$
where m_microsomes_ is 61, 45 and 40 mg, and represents the mass of the microsome per grams of liver (g_liver_) for rat, mouse and human, respectively [23,24,25,26,27]. The m_liver_ is 40, 87.5 and 25.7 g/kg and represents the mass of liver contained in each kg of body weight (kg_per body weight_) for rat, mouse and human, respectively [23,28].

The acquired chromatograms and mass spectra were analyzed using the Insight Explore software version 1.0.0.0 (Shimadzu, Kyoto, Japan). Ions with *m*/*z* corresponding to compounds with theoretical structures compatible with the main phase I metabolism reactions were investigated. In order to identify NEP and its metabolites, ions with a maximum mass error of 5 ppm for precursor ions and 10 ppm for their products were considered. To characterize the possible structures of the metabolites, only spectra with at least two ions consistent with their fragmentation mechanisms were considered, in addition to the identification of the precursor itself. Furthermore, peaks with spectra that did not meet these acceptance criteria were excluded from characterizing a metabolite.

## 3. Results

### 3.1. Metabolic Stability of NEP in RLM, MLM and HLM

NEP metabolic stability was assessed by incubating 2 µM NEP and 0.5 mg/mL RLM, MLM and HLM. Calculation of the in vitro NEP metabolic stability was achieved by plotting the remaining percentage of NEP on the y-axis and the incubation time on the x-axis, (Figure 1a). Additionally, the slope of the linear portion of the natural logarithm (ln) of the remaining percentage of NEP (y-axis) versus the incubation time (x-axis) (Figure 1b) is the NEP metabolism rate constant (k) (Table 1). The linear regression equation and the coefficient of determination (r^2^) of the linear portion of this graph also provide important information.

Through the equations described in Section 2.5, parameters such as the in vitro elimination half-life (t_1/2_), in vitro intrinsic clearance (Cl _int, in vitro_) and in vivo intrinsic clearance (Cl _int, in vivo_) were calculated for all the microsomal models and are displayed in Table 2.

### 3.2. Identification of NEP Metabolites in RLM, MLM and HLM

NEP and twelve metabolites were identified (Figure 2, Figure 3 and Figure 4 and Table 3), with eight produced by phase I reactions and four by phase II reactions. All the metabolites were identified following the criteria described in Methods 4.5, except for metabolites 6, 9 and 12. Metabolite 6 had one fragment with a mass error greater than 10 ppm (−15.5 ppm), while metabolites 9 (24.7 and 12.1 ppm) and 12 (11.9 and 25.5 ppm) had two fragments with larger mass errors.

Phase I metabolites were produced by (1) N-dealkylation, (2) beta-ketone reduction, (3) aromatic hydroxylation and (4) aliphatic hydroxylation. Metabolite 1 (M1) was generated by N-dealkylation and primarily identified in the RLM and MLM models, in contrast to a reduced relative abundance in HLM. M2 was created by a beta-ketone reduction and was prominent in RLM and HLM. In fact, it was the major metabolite in HLM, increasing from 6% at 15 min, 13% at 30 min, and almost 20% after 60 min incubation. In contrast, M2’s abundance in MLM demonstrated a lower production rate, with 0.4%, 0.9% and 7% after 15, 30 and 60 min, respectively, demonstrating an interspecies difference in its formation. M3 resulted from aromatic ring hydroxylation and was only found following RLM and MLM incubations. Interestingly, the M3-extracted chromatogram peak potentially suggested the presence of two other coeluting substances, possibly position isomers for aromatic hydroxylation. Considering this hydroxylation reaction, the formation of 3,4 or 2,3 arene oxides could result in 2′-, 3′- and 4′-hydroxy-NEP, even though 4′-hydroxylation is usually favored [29]. M6 was produced by an aliphatic chain hydroxylation reaction, but although identified in all liver microsome models, its formation was less than 4% in all. The following are considered secondary metabolites, because they are produced by two succeeding enzymatic reactions. M4 was formed following N-dealkylation plus a beta-ketone reduction, and M5 after N-dealkylation plus an aromatic hydroxylation. M4 was identified in substantial amounts only following RLM incubation, while M5 was only found in the RLM and MLM models. M7 and M8 were generated by an N-dealkylation plus an aliphatic hydroxylation, and a beta-ketone reduction added to an aromatic hydroxylation, respectively. Both only occurred in RLM.

Four glucuronide metabolites were identified through a sequential phase I and phase II approach. M9 was generated by an aromatic hydroxylation followed by O-glucuronidation on the newly added phenolic hydroxyl. M10 was created by hydroxylation in the aliphatic side chain followed by O-glucuronidation. M11 was synthesized by N-dealkylation, aromatic hydroxylation and O-glucuronidation of the phenolic hydroxyl and M12 by N-dealkylation, hydroxylation in the aliphatic side chain and O-glucuronidation. When evaluating only phase II metabolism, none of the metabolites but only the parent NEP was identified.

Based on the structures of the metabolites identified in this study, it was possible to devise a metabolic pathway (Figure 5). The metabolites’ production in all the microsomal models considered only phase I, and phase I followed by phase II reactions, as seen in Figure 6.

**Figure 2 pharmaceutics-16-00257-f002:**
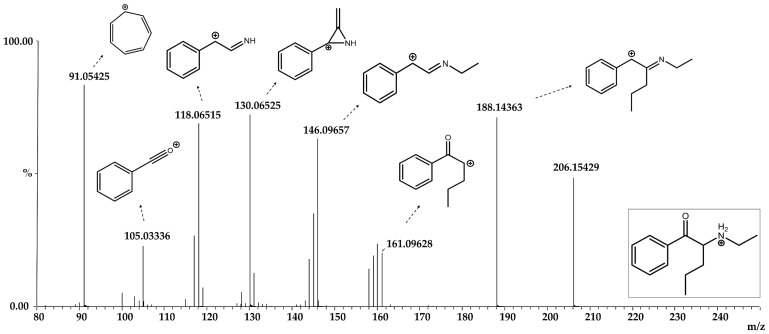
Mass spectra of N-ethyl pentedrone.

**Figure 3 pharmaceutics-16-00257-f003:**
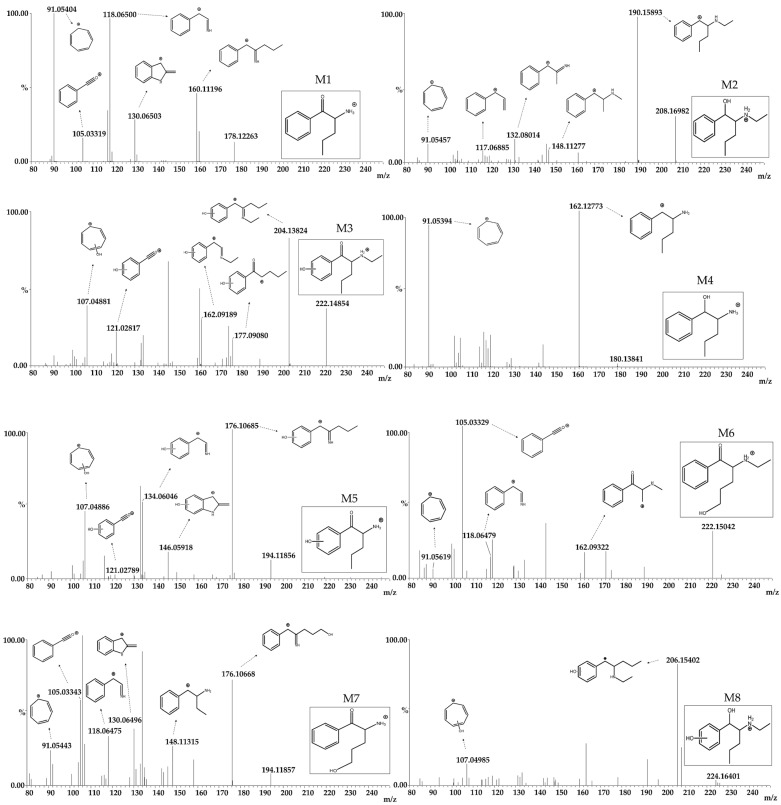
Mass spectra of N-ethyl pentedrone metabolites identified after phase I metabolism (M1–M8) and after phase I + II metabolism (M9–M12).

**Figure 4 pharmaceutics-16-00257-f004:**
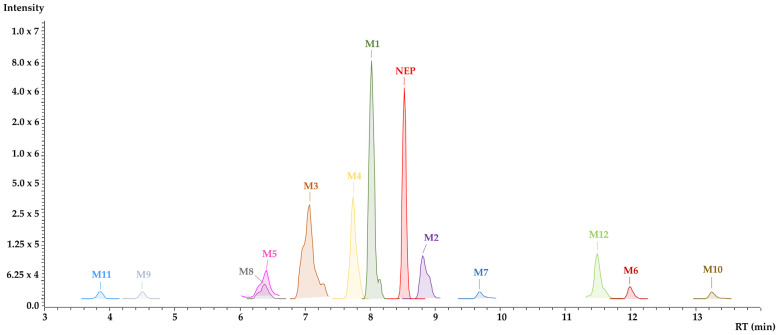
Extracted chromatogram of N-ethyl pentedrone and metabolites identified using the value of their theoretical exact mass following 60 min incubation in rat liver microsomes (RLM).

**Figure 5 pharmaceutics-16-00257-f005:**
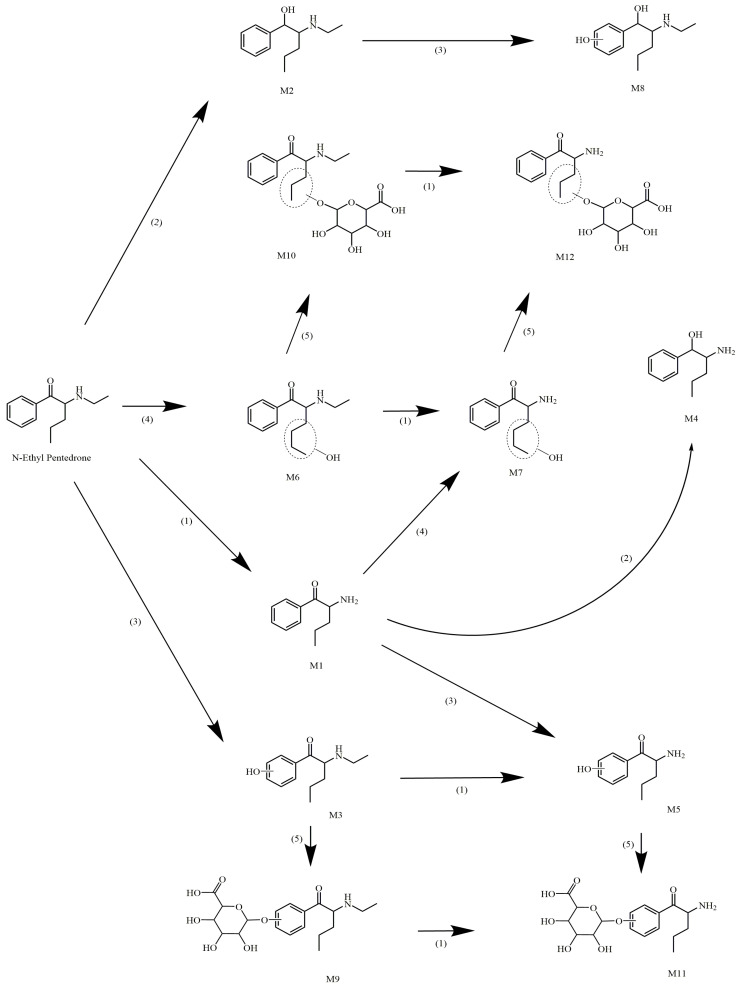
Metabolic pathway of NEP. (1) N-dealkylation, (2) beta-ketone reduction, (3) aromatic hydroxylation, (4) aliphatic hydroxylation, and (5) O-glucuronidation.

**Figure 6 pharmaceutics-16-00257-f006:**
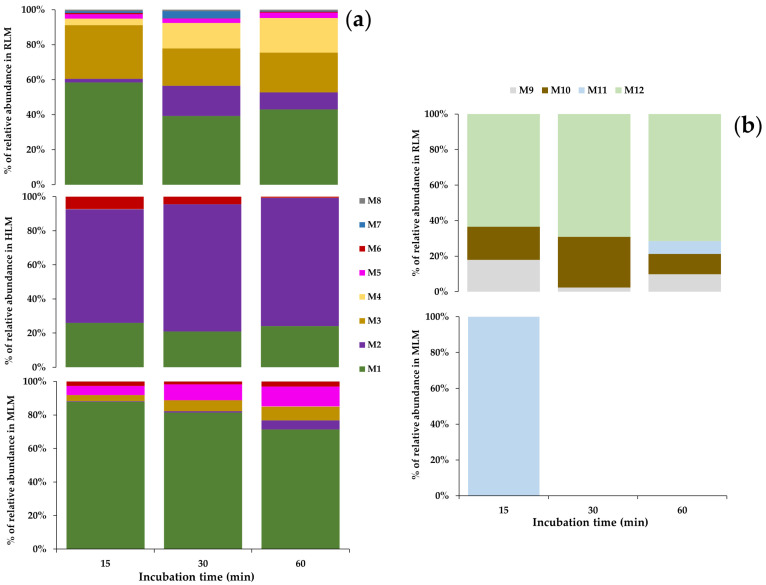
Formation of NEP metabolites in RLM, MLM and HLM considering (**a**) only phase I metabolism and (**b**) phase I metabolism followed by phase II reactions.

## 4. Discussion

A comparison of the experimentally determined kinetic parameters allows us to infer differences in metabolic stability in each microsomal model. The t_1/2_ reflects the time in which 50% of the initial amount of the drug is metabolized, and the intrinsic clearance describes the rate of elimination of the substance. A t_1/2_ = 12.1 min and Cl _int, in vitro_ = 229 μL/min/mg for NEP in the RLM model demonstrates rapid metabolism. Slower metabolism rates were observed for the MLM and HLM models, with t_1/2_ = 187 and 770 min, respectively, and Cl _int, in vitro_ = 14.8 and 3.6 μL/min/mg, respectively. Cl _int, in vivo_ was estimated utilizing allometric scaling for the three species [30,31]. Similar to the in vitro calculations, we found higher Cl _int, in vivo_ for the RLM model (128 mL/min/kg), showing a two-fold increase when compared to MLM (58.3 mL/min/kg) and almost 40-fold compared to HLM (3.7 mL/min/kg).

The metabolic profile described by the HLM model demonstrates that NEP is slowly metabolized by the human microsomal enzymatic system. These results are in agreement with an intoxication case related to NEP in Belgium, where 17 h after the patient’s admission, plasma concentrations had not substantially decreased [6]. The present findings corroborate this case report, but also other studies showing synthetic cathinones with a lateral alkyl chain moiety had increased metabolic stability [32].

NEP metabolism rates varied by species, but metabolites were also produced in different proportions in the different liver microsomes, as shown in Figure 6. Considering only phase I metabolism, N-dealkylation was the primary metabolic reaction in the RLM model producing M1 in large quantities. M3 was produced by aromatic hydroxylation after 15 min incubation, as well as metabolites created by the beta-ketone reduction reaction (M2) and N-dealkylation associated with a reduction in beta-ketone (M4) from 30 min onwards. The metabolites formed by phase I reactions became substrates for conjugation enzymes responsible for phase II metabolism, mainly after aromatic and aliphatic hydroxylations. In RLM, three glucuronide-conjugated metabolites were detected within 15 min (M9, M10 and M12); however, M11 was only identified after 60 min.

In HLM phase I metabolism, few metabolites formed by aliphatic and aromatic hydroxylation were observed, as was seen in RLM. N-dealkylation products achieved only about 6% abundance after 60 min. As demonstrated in the metabolic stability experiment, NEP metabolism in HLM occurs at a low rate because reactions contributing greatly to NEP decreases in RLM (N-dealkylation and aromatic hydroxylation) do not occur extensively in the HLM model. The main metabolite in HLM is M2, followed by M1; however, the rates of production were much lower than those observed in the RLM model. No conjugated metabolites were found in HLM when evaluating phase I followed by phase II reactions and only phase II metabolism. The absence of these metabolites may be due to the poor generation of hydroxylated products by phase I metabolism. In fact, other publications regarding the phase II metabolism of synthetic cathinones already reported a limited formation of such conjugated metabolites in HLM and rat hepatocytes [33,34].

The rates of metabolite production in the MLM model regarding only phase I metabolism were similar to the RLM pattern, with a remarkable importance of M1, M2 and M3. Differences between these two models included the lower formation of M4 and a higher proportion of M5 in the MLM model. One phase II metabolite (M11) was identified in low amounts just after 15 min incubation. The lower diversity of the conjugated metabolites found during MLM phase I followed by phase II metabolism could be explained by the minimal formation of NEP phase I-derivatives capable of phase II conjugation. The presence of M11 in this model occurred by an O-glucuronidation of M5. 

Interspecies differences in the rates of metabolite production could be explained by differential CYP450 isoform compositions, expression and catalytic activities. Alterations in subfamilies such as CYP1A2, -2C and -3A are mainly associated with variability in the metabolic stability of drugs, as there are substantial differences within the species evaluated in this study [35]. These data show that rat and mouse are not good animal models for the prediction of human in vivo NEP metabolism. Thus, studies using other liver microsomal species, such as dog or pig, and/or additional in vivo experiments, should be performed to provide a better correlation with human NEP metabolism. Furthermore, as many a posteriori toxicological studies employ in vivo assays, choosing the best correlate model to humans can provide a more confident comparability.

We infer that phase II metabolites of NEP can only be generated after phase I metabolism. The set of phase I reactions act as functionalization steps, providing reactive moieties that can conjugate with UDGPA biomolecules. For NEP, aromatic and aliphatic hydroxylation preceded the formation of glucuronide metabolites. Sulfate and methyl NEP conjugates were not observed.

## 5. Conclusions

We presented an evaluation of metabolic stability through three different microsomal models of an NPS seized and detected worldwide. Lower t_1/2_, associated with higher Cl _int, in vitro_ and Cl _int, in vivo_ in HLM, demonstrated a lasting stability of NEP in this model when compared to RLM and MLM. These data provide important data for clinicians receiving NEP-related intoxications. Furthermore, twelve metabolites were identified following phase I and phase II reactions. The identification of NEP metabolites provides critical information for the detection and consumption control of this substance. Interestingly, our study also demonstrated the importance of choosing the suitable model for assessing metabolic parameters of drugs, in which we observed important differences among the evaluated models.

## Figures and Tables

**Figure 1 pharmaceutics-16-00257-f001:**
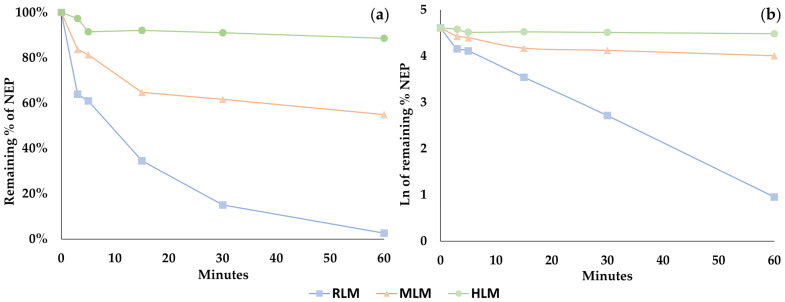
(**a**) Metabolic stability of N-ethyl pentedrone (NEP) in rat (RLM), mouse (MLM) and human (HLM), (**b**) and the linear plot using the natural logarithm (ln) of the remaining percentage of NEP.

**Table 1 pharmaceutics-16-00257-t001:** N-ethyl pentedrone (NEP) metabolic stability in rat (blue), mouse (orange) and human (green) liver microsomes.

RLM	Time (min)	Average Area	RSD (%)	Z (%)	ln Z	Linear Regression Equation and Analytical Parameters
	0	1,094,637	13.7	100	4.605	
	3	700,048	1.1	64	4.158	
	**5**	**667,338**	**7.2**	**61**	**4.110**	y = −0.0573 + 4.4068
	**15**	**377,918**	**4.3**	**35**	**3.542**	
	**30**	**164,819**	**15.4**	**15**	**2.712**	r^2^ = 1.000
	**60**	**28,459**	**9.9**	**3**	**0.955**	Slope (k) = 0.0573
**MLM**	**Time (min)**	**Average Area**	**RSD (%)**	**Z (%)**	**ln Z**	**Linear Regression Equation** **and Analytical Parameters**
	0	1,398,338	2.8	100	4.605	
	3	1,170,931	8.9	84	4.428	
	5	1,137,263	1.4	81	4.399	y = −0.0037 + 4.2284
	**15**	**905,606**	**3.9**	**65**	**4.171**	
	**30**	**862,389**	**1.0**	**62**	**4.122**	r^2^ = 1.000
	**60**	**768,422**	**2.5**	**55**	**4.006**	Slope (k) = 0.0037
**HLM**	**Time (min)**	**Average Area**	**RSD (%)**	**Z (%)**	**ln Z**	**Linear Regression Equation** **and Analytical Parameters**
	0	1,247,875	1.1	100	4.605	
	3	1,215,356	1.0	97	4.579	
	5	1,142,124	0.8	92	4.517	y = −0.0009 + 4.5367
	**15**	**1,149,615**	**0.6**	**92**	**4.523**	
	**30**	**1,135,939**	**1.1**	**91**	**4.511**	r^2^ = 1.000
	**60**	**1,105,557**	**0.7**	**89**	**4.484**	Slope (k) = 0.0009

Areas displayed are the average of two replicates and the linear range of the decay curve plotted (ln Z within time) in bold. RSD, the relative standard deviation; Z, the remaining NEP percentage.

**Table 2 pharmaceutics-16-00257-t002:** Stability parameters of N-ethyl pentedrone in rat, mouse and human liver microsomes.

Parameter	Species		
	Rat	Mouse	Human
In vitro elimination half-life (min)	12.1	187	770
In vitro intrinsic clearance (µL/min/mg)	229	14.8	3.6
In vitro intrinsic clearance (mL/min/kg)	128	58.3	3.7

**Table 3 pharmaceutics-16-00257-t003:** N-ethyl pentedrone and metabolite data after incubation in rat (RLM), mouse (MLM) and human (HLM) liver microsomes.

Microsomal Model	Molecule	Metabolism Reaction	Fragment	Molecular Formula	Theoretical Exact Mass [M + H]^+^	Measured Exact Mass [M + H]^+^	Mass Error (ppm)	Retention Time (min)
RLM, HLM and MLM	NEP	-	-	C_13_H_20_NO	206.15449	206.15429	0.97	8.55
			F1	C_13_H_18_N	188.14392	188.14363	1.56	
			F2	C_11_H_13_O	161.09664	161.09628	2.23	
			F3	C_10_H_12_N	146.09697	146.09657	2.77	
			F4	C_9_H_8_N	130.06567	130.06525	3.26	
			F5	C_8_H_8_N	118.06567	118.06515	4.44	
			F6	C_7_H_5_O	105.03404	105.03336	6.47	
			F7	C_7_H_7_	91.05478	91.05425	5.77	
RLM, HLM and MLM	M1	N-dealkylation	-	C_11_H_16_NO	178.12319	178.12263	3.14	8.15
			F8	C_11_H_14_N	160.11262	160.11196	4.14	
			F4	C_9_H_8_N	130.06567	130.06503	4.95	
			F5	C_8_H_8_N	118.06567	118.06500	5.71	
			F6	C_7_H_5_O	105.03404	105.03319	8.09	
			F7	C_7_H_7_	91.05478	91.05404	8.07	
RLM, HLM and MLM	M2	Beta-ketone reduction	-	C_13_H_22_NO	208.17014	208.16982	1.53	8.79
			F9	C_13_H_20_N	190.15957	190.15893	3.39	
			F10	C_10_H_14_N	148.11262	148.11277	−0.99	
			F11	C_9_H_10_N	132.08132	132.08014	8.96	
			F7	C_7_H_7_	91.05478	91.05457	2.25	
RLM and MLM	M3	Aromatic hydroxylation	-	C_13_H_20_NO_2_	222.14940	222.14854	3.89	7.06
			F12	C_13_H_18_NO	204.13884	204.13824	2.93	
			F13	C_11_H_13_O_2_	177.09156	177.09080	4.26	
			F14	C_10_H_12_NO	162.09189	162.09189	−0.01	
			F15	C_7_H_5_O_2_	121.02896	121.02817	6.49	
			F16	C_7_H_7_O	107.04969	107.04881	8.22	
RLM, HLM and MLM	M4	N-dealkylation + Beta-ketone reduction	-	C_11_H_18_NO	180.13884	180.13841	2.38	7.70
			F17	C_11_H_16_N	162.12827	162.12773	3.36	
			F7	C_7_H_7_	91.05478	91.05394	9.17	
RLM and MLM	M5	N-dealkylation + Aromatic hydroxylation	-	C_11_H_16_NO_2_	194.11810	194.11856	−2.35	6.31
			F18	C_11_H_14_NO	176.10754	176.10685	3.91	
			F19	C_9_H_8_NO	146.06059	146.05918	9.65	
			F20	C_8_H_8_NO	134.06059	134.06046	0.96	
			F15	C_7_H_5_O_2_	121.02896	121.02789	8.80	
			F16	C_7_H_7_O	107.04969	107.04886	7.75	
RLM, HLM and MLM	M6	Aliphatic hydroxylation	-	C_13_H_20_NO_2_	222.14940	222.15042	−4.57	11.98
			F14	C_10_H_12_NO	162.09189	162.09322	−8.21	
			F5	C_8_H_8_N	118.06567	118.06479	7.49	
			F6	C_7_H_5_O	105.03404	105.03329	7.14	
			F7	C_7_H_7_	91.05478	91.05619	−15.54	
RLM and MLM	M7	N-dealkylation + Aliphatic hydroxylation	-	C_11_H_16_NO_2_	194.11810	194.11857	−2.40	9.65
			F18	C_11_H_14_NO	176.10754	176.10668	4.88	
			F10	C_10_H_14_N	148.11262	148.11315	−3.56	
			F4	C_9_H_8_N	130.06567	130.06496	5.49	
			F5	C_8_H_8_N	118.06567	118.06475	7.83	
			F6	C_7_H_5_O	105.03404	105.03343	5.81	
			F7	C_7_H_7_	91.05478	91.05443	3.79	
RLM and MLM	M8	Beta-ketone reduction + Aromatic hydroxylation	-	C_13_H_22_NO_2_	224.16505	224.16401	4.66	6.41
			F21	C_13_H_20_NO	206.15449	206.15402	2.27	
			F16	C_7_H_7_O	107.04969	107.04985	−1.49	
RLM	M9	Aromatic hydroxylation + O-glucuronidation	-	C_19_H_28_NO_8_	398.18149	398.18273	−3.10	4.52
			F22	C_13_H_20_NO_2_	222.14940	222.14979	−1.74	
			F12	C_13_H_18_NO	204.13884	204.13965	−3.97	
			F13	C_11_H_13_O_2_	177.09156	177.09296	−7.93	
			F14	C_10_H_12_NO	162.09189	162.09055	8.26	
			F15	C_7_H_5_O_2_	121.02896	121.02597	24.66	
			F16	C_7_H_7_O	107.04969	107.04839	12.14	
RLM	M10	Aliphatic hydroxylation + O-glucuronidation	-	C_19_H_28_NO_8_	398.18149	398.18242	−2.33	13.24
			F23	C_13_H_20_NO_2_	222.14940	222.14962	−0.97	
			F14	C_10_H_12_NO	162.09189	162.09204	−0.93	
			F5	C_8_H_8_N	118.06567	118.06545	1.90	
			F6	C_7_H_5_O	105.03404	105.03371	3.14	
RLM	M11	N-dealkylation + Aromatic hydroxylation + O-glucuronidation	-	C_17_H_24_NO_8_	370.15019	370.15092	−1.96	3.84
			F24	C_11_H_16_NO_2_	194.11810	194.11769	2.13	
			F18	C_11_H_14_NO	176.10754	176.10760	−0.35	
			F19	C_9_H_8_NO	146.06059	146.06104	−3.09	
			F20	C_8_H_8_NO	134.06059	134.06069	−0.75	
			F15	C_7_H_5_O_2_	121.02896	121.02880	1.28	
			F16	C_7_H_7_O	107.04969	107.04995	−2.43	
RLM	M12	N-dealkylation + Aliphatic hydroxylation + O-glucuronidation	-	C_17_H_24_NO_8_	370.15019	370.15112	−2.50	11.49
			F25	C_11_H_16_NO_2_	194.11810	194.11793	0.90	
			F18	C_11_H_14_NO	176.10754	176.10682	4.08	
			F10	C_10_H_14_N	148.11262	148.11320	−3.90	
			F4	C_9_H_8_N	130.06567	130.06455	8.64	
			F5	C_8_H_8_N	118.06567	118.06563	0.37	
			F6	C_7_H_5_O	105.03404	105.03279	11.90	
			F7	C_7_H_7_	91.05478	91.05245	25.53	

## Data Availability

The data presented in this study are available in this article.

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
