# Peer review of "Metabolic Stability and Metabolite Identification of N-Ethyl Pentedrone Using Rat, Mouse and Human Liver Microsomes"

_pharmaceutics, 2024, doi:10.3390/pharmaceutics16020257_

Round 1

Reviewer 1 Report

Comments and Suggestions for Authors

1.     The authors are suggested to describe the positive control compounds been used.

2.     The negative control should be: incubation of compound in the buffer, and incubation with microsomes only. The incubation of substance in the absence of cofactor solutions cannot exclude metabolism that does not require cofactors.

3.     The structure of M8, M10 and M12 should be draw in Markush structure.

4.     The figure 6 is unclear: why the formation of Phase I and Phase II metabolites is less than Phase I metabolites. I assume the authors means the metabolites from Phase II metabolism followed by Phase I metabolism is less than Phase I only metabolism. If so, this information could be described in the manuscript.

5.     The quality of the MS/MS should be improved to increase the readability.

Comments on the Quality of English Language

The manuscript is suggested to be proof-read by native English speakers.

Author Response

  1. The authors are suggested to describe the positive control compounds been used.

As suggested, we added the use of dapaconazole as the positive control in our incubations that was evaluated under the same conditions as employed for NEP. Descriptions of the positive control were cited at page 2, lines 80-83 and at page 3, lines 112-113, as follows:

  • Page 2, lines 80-83: Dapaconazole, the positive control for the experiment, was supplied by Biolab Farmacêutica Ltda. (São Paulo, Brazil) and a stock solution prepared at 1 mg/mL (2.4 mM) in methanol. Dapaconazole working solution of 40 μM was prepared from the dapaconazol stock solution in methanol.
  • Page 3, lines 112-113: Positive controls were prepared by incubating dapaconazole under the same condition as NEP (incubation concentration 2 μM).
  1. The negative control should be: incubation of compound in the buffer, and incubation with microsomes only. The incubation of substance in the absence of cofactor solutions cannot exclude metabolism that does not require cofactors.

We failed to make it clear that we not only incubated NEP in the absence of the cofactors, as previously described, but also incubated NEP only in buffer, ensuring a reliable NEP decay curve. As described in “The Conduct of Drug Metabolism Studies Considered Good Practice (II): In Vitro Experiments” by Jia and Liu, negative controls prepared in the absence of microsomes and cofactors allow the determination of drug stability in PBS, avoiding protein and energy dependent reactions. This information was added in the text at page 3, line 113-115, as follows: Negative controls were prepared by incubating NEP in buffer solution, in the absence of microsome and cofactor solutions.

  1. The structure of M8, M10 and M12 should be draw in Markush structure.

As suggested, Figure 5 was revised with Markush structures.

  1. The figure 6 is unclear: why the formation of Phase I and Phase II metabolites is less than Phase I metabolites. I assume the authors means the metabolites from Phase II metabolism followed by Phase I metabolism is less than Phase I only metabolism. If so, this information could be described in the manuscript.

As suggested, phase I metabolites followed by phase II reactions were less abundant than only phase I metabolites. We improved the description of such findings in the manuscript. The alterations were made at page 8, lines 264-265, page 14, lines 286-287, page 15, line 324 and 334, as follows:

  • Page 8, lines 264-265: Metabolites’ production in all microsomal models considered only phase I, and phase I followed by phase II reactions as seen in Figure 6.
  • Page 14, lines 286-287: Figure 6. Formation of NEP metabolites in RLM, MLM and HLM considering (a) only phase I metabolism and (b) phase I metabolism followed by phase II reactions.
  • Page 15, line 324: No conjugated metabolites were found in HLM when evaluating phase I followed by phase II reactions and only phase II metabolism.
  • Page 15, line 334: The lower diversity of conjugated metabolites found during MLM phase I followed by phase II metabolism could be explained by minimal formation of NEP phase I-derivatives capable of phase II conjugation.
  1. The quality of the MS/MS should be improved to increase the readability.

As suggested, we improved the resolution of all MS/MS spectra in the manuscript.

Comments on the Quality of English Language: The manuscript is suggested to be proof-read by native English speakers.

The manuscript was written and fully reviewed by a co-author that is a native English speaker.

Reviewer 2 Report

Comments and Suggestions for Authors

The manuscript of  Barcia de Godoi et al. describes the in vitro metabolism of a cathinone drug N-ethyl-pentedrone, (NEP) in rat, mouse and human liver microsomes. The authors provide preliminary pharmacokinetic parameters and the identification of 12 phase I and phase II metabolites.

The paper is well written and easy to read. The study is correctly conceived. However there are a number of small errors in the way the work is described.

Below are a few comments on what could be improved.

Experimental part : For me the incubation conditions are not very clear. I do not understand in reading the manuscript how the experiment was exactly performed : thus the following remarks :

            Page 2 third paragraph : 1mg/mL of NEP is about 5 microM/mL or 5 mM. 10 microL / 200 microL incubation is about 250 microM. What is the final concentration of NEP in the incubation. You say that the mother solution of NEP is 1mg/mL that is about 5 mM. What dilution do you use for pipetting the 10 microL per tube? I cannot see from your text  the final concentration.

            I really think your incubation method is not well described. Do you use 100 microl of a solution of NEP plus Generating System and then add 100 microl microsomes so that at the end you have 200 microL containing 2 microM NEP and 0.5 mg protein/mL? Then your sentence should make it clear.

If I understand your final conc of NADPH is 0.25 mM . Most often, people use 0.6-1 mM. Thus your NADPH is suboptimal (but probably sufficient since the Km for NADPH cytochrome P450 reductase is below 0.1 mM).

            For the metabolite identification experiment :  Is 37 microM the final concentration in the incubation? Why chose this value and not 50 or 100 microM. Since you have a dilution when you add the microsomes you must state the final concentration. Otherwise one cannot reproduce your experiment.

            Another remark ; do you know the amount of cytochrome P450/ mg protein  in your  microsomes. This can be easily measured using the method of Omura and Sato by UV visible spectrometry. (CO bonding spectrum). I think, suppliers of human liver microsomes generally give you this parameter (at least Gentest and Xenotech and BioIVT). I don't know about Sigma. If you have these values give them.

            Page 5 : figure 1 is fuzzy. Could you use high resolution pictures. 

            Page 7 : Kinetics : If I understand the kinetic elimination study was done at 2 microM. If the Km of some P450 enzymes is largely above 50 microM, the elimination rate would be much underestimated by performing the study at 2 microM

Do you know what is the possible cell concentration (and plasma concentration) of in vivo intoxication with this drug in Human?

            Page 7 fig 2 : Again the figure is fuzzy. Correct this. 

About the fragmentations shown, I don't know your software. But in general the 91 ion is the tropylium ion (rearrangement of the phenylmethylenium ion, tropylenium is more stable). Your sofware should predict it since it is in all the textbooks on mass spectrometry.

For M3 the formation of the 91 ion (C7H7+) is difficult to imagine. But its a fact. I suppose 107 is Hydroxy-tropylium. But losing an oxygen seem difficult.

            Figure 4 (it is also fuzzy) On this figure, the M3 peak looks strange. There are 2 shoulders. It may contain several isomers. (2 or 3). Similarly M2 has one shoulder and also M1 has one shoulder. The parent compound and other metabolites peaks are symmetrical. You could have aromatic hydroxylation into for instance 3 and 4 hydroxylation through a 3,4 arene oxide and 2 and 3 hydroxylation through a 2,3 arene oxide. Thus 2, 3 and 4 OH derivatives. (are there similar finding in the literature for small aromatic alky amines).

            Page 14 : Figure 6 is difficult to read. It cannot be understood by by real colorblind.

I think you should use a more classical representation. Perhaps graphs (linear plots) showing each metabolite concentation in function of time for each type of microsomes. (The percent would not be present but more the actual rate.)

If you use conc /mg protein, you could change the title to : Consumption (or disappearance) of NEP and formation of NEP metabolites : a) phase I metabolites, b) phase I + phase II metabolites.

You must try what is best. Perhaps do not show the disappearance of NEP  so that you can use a bigger scale for metabolites.

One more point : You did not try S9 supernatant supplied with PAPS that could make sulfate conjugates that are also possible metabolites in vivo. For instance the sulfate is the leading metabolite of paracetamol at low concentration (10 mg /kg in rat). The glucuronide being found for 10 time higher concentrations. You perhaps could check S9 plus NADPH plus PAPS for the 3 species.

In conclusion, the paper is interesting, brings a number of new data and I think it should be acceptable after minor corrections.

Comments on the Quality of English Language

The langage is correct. 

Author Response

The manuscript of Barcia de Godoi et al. describes the in vitro metabolism of a cathinone drug N-ethyl-pentedrone, (NEP) in rat, mouse and human liver microsomes. The authors provide preliminary pharmacokinetic parameters and the identification of 12 phase I and phase II metabolites. The paper is well written and easy to read. The study is correctly conceived.

However, there are a number of small errors in the way the work is described. Below are a few comments on what could be improved. Experimental part: For me the incubation conditions are not very clear. I do not understand in reading the manuscript how the experiment was exactly performed: thus, the following remarks:

  1. Page 2 third paragraph: 1mg/mL of NEP is about 5 microM/mL or 5 mM. 10 microL / 200 microL incubation is about 250 microM. What is the final concentration of NEP in the incubation. You say that the mother solution of NEP is 1mg/mL that is about 5 mM. What dilution do you use for pipetting the 10 microL per tube? I cannot see from your text the final concentration.

As suggested, we improved our description of the concentration of NEP. We prepared NEP hydrochloride at 1 mg/mL from Cayman Chemical equivalent to a concentration of 849 μg/mL free base NEP or about 4.1 mM. This information is included in the manuscript at page 2, lines 76-80. Next, we diluted the NEP stock (849 μg/mL) solution to a working solution of 40 μM in methanol, by adding 10 μL of NEP 4.1 mM to 1 mL of methanol, followed by thorough mixing. 10 μL of this NEP 40 μM solution was pipetted into tubes and dried under nitrogen. The NEP was resuspended in 100 μL of a solution containing the cofactors in PBS resulting in 4 μM NEP. Finally, 100 μL microsomes were added to start the reaction, yielding a final 2 μM NEP concentration. The same procedure was performed for metabolites identification incubations, but making a working solution at 732 μM in methanol. NEP 732 μM was prepared adding 178.5 μL of NEP 4.1 mM to 1 mL of methanol. We revised the manuscript to better explain the experiment (page 2, lines 76-80; page 3, lines 98-107, as follows:

  • Page 2, lines 76-80: NEP hydrochloride reference material (98% purity) was purchased from Cayman Chemical (Ann Arbor, MI, USA) and a stock solution prepared at 1 mg/mL (4.1 mM free base) in methanol. NEP working solutions of 40 and 732 μM were also prepared by appropriate dilution of NEP stock solution in methanol.
  • Page 3, line 98-107: Ten microliters of 40 µM NEP was added to 1.5 mL propylene tubes and dried under nitrogen. Then, 100 μL of NADPH-regenerating system, containing 1.1 mM NADP+, 10 mM glucose-6-phosphate, 1 U/mL glucose-6-phosphate dehydrogenase, 5 mM sodium citrate and 66 mM magnesium chloride in 100 mM phosphate buffer, pH 7.4 was added to the tubes. The solutions were then pre-incubated for 5 min in a MTC 100 thermo shaker incubator (Miulab, Zhejiang, China) at 300 rpm and 37°C. To start the reactions, aliquots of 100 μL of RLM, MLM and HLM at 1 mg/mL of protein were added to the pre-incubated shaking tubes yielding NEP and protein final concentrations of 2 μM and 0.5 mg/mL, respectively. After 0, 3, 5, 15, 30 and 60 min, the reactions were stopped by adding 400 μL ice-cold acetonitrile to the medium.
  1. I really think your incubation method is not well described. Do you use 100 microl of a solution of NEP plus Generating System and then add 100 microl microsomes so that at the end you have 200 microL containing 2 microM NEP and 0.5 mg protein/mL? Then your sentence should make it clear.

As suggested, we improved this sentence to better describe our incubation procedures at page 2, line 76-80, and page 3, lines 98-107, as follows:

  • Page 2, line 76-80: NEP hydrochloride reference material (98% purity) was purchased from Cayman Chemical (Ann Arbor, MI, USA) and a stock solution prepared at 1 mg/mL (4.1 mM free base) in methanol. NEP working solutions of 40 and 732 μM were also prepared by appropriate dilution of NEP stock solution in methanol.
  • Page 3, lines 98-107: Ten microliters of 40 µM NEP was added to 1.5 mL propylene tubes and dried under nitrogen. Then, 100 μL of NADPH-regenerating system, containing 1.1 mM NADP+, 10 mM glucose-6-phosphate, 1 U/mL glucose-6-phosphate dehydrogenase, 5 mM sodium citrate and 66 mM magnesium chloride in 100 mM phosphate buffer, pH 7.4 was added to the tubes. The solutions were then pre-incubated for 5 min in a MTC 100 thermo shaker incubator (Miulab, Zhejiang, China) at 300 rpm and 37°C. To start the reactions, aliquots of 100 μL of RLM, MLM and HLM at 1 mg/mL of protein were added to the pre-incubated shaking tubes yielding NEP and protein final concentrations of 2 μM and 0.5 mg/mL, respectively. After 0, 3, 5, 15, 30 and 60 min, the reactions were stopped by adding 400 μL ice-cold acetonitrile to the medium.
  1. If I understand your final conc of NADPH is 0.25 mM. Most often, people use 0.6-1 mM. Thus your NADPH is suboptimal (but probably sufficient since the Km for NADPH cytochrome P450 reductase is below 0.1 mM).

We inappropriately described the final cofactors concentration in the manuscript. The final concentration is 0.55 mM NADPH. Page 3, line 100: Then, 100 μL of NADPH-regenerating system, containing 1.1 mM NADP+, 10 mM glucose-6-phosphate, 1 U/mL glucose-6-phosphate dehydrogenase, 5 mM sodium citrate and 66 mM magnesium chloride in 100 mM phosphate buffer, pH 7.4 was added to the tubes.

  1. For the metabolite identification experiment:  Is 37 microM the final concentration in the incubation? Why chose this value and not 50 or 100 microM. Since you have a dilution when you add the microsomes you must state the final concentration. Otherwise one cannot reproduce your experiment.

We based our incubation procedures for metabolite identification on a 2022 Yeh and Wang study (https://pubmed.ncbi.nlm.nih.gov/35332690/), in which they employed an incubation concentration of 8.3 μg/mL of another synthetic cathinone, Eutylone. We followed aspects of this study because they also evaluated the formation ratio of metabolites as a function of time. Applying similar values to our experiments, we prepared a NEP plus cofactors solution of 15 μg/mL (73 μM) and then incubated 100 μL of this solution with 100 μL microsomes (achieving 7.5 μg/mL, a similar concentration used by Yeh and Wang). Thus, the final concentration of 7.5 μg/mL NEP corresponds to 37 μM. As suggested, we revised the Materials and Methods to better clarify the incubations, at page 2, lines 78-80, page 3, lines 119-128 and 133-141, and page 4, line 142, as follows:

  • Page 2, lines 78-80: NEP working solutions of 40 and 732 μM were also prepared by appropriate dilution of NEP stock solution in methanol.
  • Page 3, lines 119-128: Ten microliters of 732 μM NEP was added to 1.5 mL propylene tubes and dried under nitrogen. Then, 100 μL of NADPH-regenerating system, containing 1.1 mM NADP+, 10 mM glucose-6-phosphate, 1 U/mL glucose-6-phosphate dehydrogenase, 5 mM sodium citrate and 66 mM magnesium chloride in 100 mM phosphate buffer, pH 7.4 was added to the tubes. The tubes were pre-incubated for 5 min in a MTC 100 thermo shaker incubator at 300 rpm and 37°C. In order to start the reactions, aliquots of 100 μL of RLM, MLM and HLM at 5 mg/mL were added in the pre-incubated shaking tubes, achieving NEP and protein final concentrations of 37 μM and 2.5 mg/mL, respectively. The metabolism reactions were stopped after 0, 15, 30, 60 min by adding 400 μL ice-cold acetonitrile to the medium.
  • Page 3, lines 133-141: Ten microliters of 732 μM NEP was added to 1.5 mL propylene tubes and dried under nitrogen. Then, 100 μL of 13.70 mM UDPGA, 0.49 mM PAPS, 3.79 mM SAM in 100 mM phosphate buffer, pH 7.4, was added to the tubes in the presence (phase I followed by phase II) and absence (only phase II) of phase I cofactors, in order to access phase II dependency of phase I metabolism. The tubes were pre-incubated for 5 min in a MTC 100 thermo shaker incubator at 300 rpm and 37°C. In order to start the reactions, aliquots of 100 μL of RLM, MLM and HLM at 5 mg/mL of protein were added in the pre-incubated shaking tubes, achieving NEP and protein final concentration of 37 μM and 2.5 mg/mL, respectively. The metabolism reactions were stopped
  • Page 4, line 142: after 0, 15, 30, 60 min by adding 400 μL ice-cold acetonitrile to the medium.
  1. Another remark; do you know the amount of cytochrome P450/ mg protein in your microsomes. This can be easily measured using the method of Omura and Sato by UV visible spectrometry. (CO bonding spectrum). I think, suppliers of human liver microsomes generally give you this parameter (at least Gentest and Xenotech and BioIVT). I don't know about Sigma. If you have these values give them.

As suggested this information was added at page 2, lines 85-86. Revisions were as follows: Pooled HLM containing 20 mg/mL microsomal proteins and 270 pmol CYP450/mg protein were obtained from Sigma-Aldrich.

  1. Page 5: figure 1 is fuzzy. Could you use high resolution pictures. 

As suggested, we improved the resolution of figure 1 to 300 dpi (4000 x 1535 pixels) and increased the type size of the axis. Figure 1 was corrected at page 5.

  1. Page 7: Kinetics: If I understand the kinetic elimination study was done at 2 microM. If the Km of some P450 enzymes is largely above 50 microM, the elimination rate would be much underestimated by performing the study at 2 microM

We followed the guidelines for drug metabolism experiments specifically for microsome incubations, “The Conduct of Drug Metabolism Studies Considered Good Practice (II): In Vitro Experiments” by Jia and Liu, which recommends: “The in vitro experimental concentration of a drug should be a little higher than the maximal blood concentration achieved in the animal study. If the concentrations have not been determined, a range of 1–10 μM of the final drug concentration can generally be used to closely mimic in vivo levels of the drug”. The authors recommended a low starting incubation concentration (1 μM) to ensure first-order metabolism generated by the bound drug-enzyme (at the low concentration (~1 μM), the drug-enzyme reaction presumably follows a first-order process, i.e., [S] <Km, and hence, the reaction rate is proportional to the drug concentration”) and also to minimize the chance of drug precipitation. As suggested, we added the guideline mentioned here in the manuscript at page 3, lines 97-98, as follows: For the metabolic stability determination, incubations followed the good practices guideline for metabolism studies [20].

  1. Do you know what is the possible cell concentration (and plasma concentration) of in vivo intoxication with this drug in Human?

NEP plasma concentrations in human intoxication cases ranged from 0.93 to 2.2 μg/mL, and in a single case, 3.1 μg/mL in serum. In all of these cases, severe intoxication symptoms were observed in patients. However, it is not possible to assign the severity of the symptoms only to NEP, because all of these users ingested other drugs.

  1. Page 7 fig 2: Again the figure is fuzzy. Correct this. 

As suggested, we improved the resolution of figure 2 to 300 dpi (8031 x 3897 pixels), increased the type size of axis and peaks and made the peaks and the molecular structures thicker. Figure 2 was corrected at page 8.

  1. About the fragmentations shown, I don't know your software. But in general the 91 ion is the tropylium ion (rearrangement of the phenylmethylenium ion, tropylenium is more stable). Your sofware should predict it since it is in all the textbooks on mass spectrometry.

As suggested, the appropriate way to report the 91 ion is as a tropylium ion. We corrected this in all described mass spectra.

  1. For M3 the formation of the 91 ion (C7H7+) is difficult to imagine. But its a fact. I suppose 107 is Hydroxy-tropylium. But losing an oxygen seem difficult.

As suggested, the 107 is the hydroxy-tropylium ion. We agree that the formation of the tropylium ion from the 107 ion is not easily achieved in our experimental conditions. For this reason, we removed the 91 ion in M3 and M5, in mass spectra (Figure 3) and Table 3.

  1. Figure 4 (it is also fuzzy) On this figure, the M3 peak looks strange. There are 2 shoulders. It may contain several isomers. (2 or 3). Similarly M2 has one shoulder and also M1 has one shoulder. The parent compound and other metabolites peaks are symmetrical. You could have aromatic hydroxylation into for instance 3 and 4 hydroxylation through a 3,4 arene oxide and 2 and 3 hydroxylation through a 2,3 arene oxide. Thus 2, 3 and 4 OH derivatives. (are there similar finding in the literature for small aromatic alky amines).

As suggested, we improved the resolution of figure 4 to 300 dpi (3963 x 1781 pixels), and increased the type size of each axis and peak. Figure 4 was corrected at page 12, line 279. We also commented on peak shapes, as suggested, at page 7, lines 242-246, as follows: Interestingly, the M3 extracted chromatogram peak suggested the presence of two other potential coeluting substances, possibly position isomers for aromatic hydroxylation. Considering this hydroxylation reaction, the formation of 3,4 or 2,3 arene oxides could result in 2’-, 3’- and 4’- hydroxy-NEP, even though 4’-hydroxylation is usually favored [29].

  1. Page 14: Figure 6 is difficult to read. It cannot be understood by by real colorblind. I think you should use a more classical representation. Perhaps graphs (linear plots) showing each metabolite concentation in function of time for each type of microsomes. (The percent would not be present but more the actual rate.)

We initially plotted a graph showing the linear plots of metabolites’ concentrations as a function of time. However, as we identified a large number of metabolites (8 by phase I metabolism and 4 by phase I followed by phase II metabolism), the graph did not clearly show the differences among formation rates. Therefore, to allow simultaneous visualization of all metabolites at different incubation times and considering all microsomal models employed, the bar charts were developed. However, we thought that if we did not show the disappearance of NEP – as you suggested in the next comment, we could improve figure visualization and understanding. We revised figure 6 at page 14, line 285.

  1. If you use conc /mg protein, you could change the title to: Consumption (or disappearance) of NEP and formation of NEP metabolites: a) phase I metabolites, b) phase I + phase II metabolites.

As suggested, we revised the title of Figure 6 following your suggestion at page 14, lines 286-287, as follows: Figure 6. Consumption of NEP and formation of NEP metabolites in RLM, MLM and HLM considering (a) only phase I metabolism and (b) phase I metabolism followed by phase II reactions.

  1. You must try what is best. Perhaps do not show the disappearance of NEP so that you can use a bigger scale for metabolites.

We thought it was a great idea. It could improve the figure visualization and made the understanding easier. So, figure 6 was reconstructed at page 14.

  1. One more point: You did not try S9 supernatant supplied with PAPS that could make sulfate conjugates that are also possible metabolites in vivo. For instance the sulfate is the leading metabolite of paracetamol at low concentration (10 mg /kg in rat). The glucuronide being found for 10 time higher concentrations. You perhaps could check S9 plus NADPH plus PAPS for the 3 species.

As suggested, we will include the S9 approach in future studies.

In conclusion, the paper is interesting, brings a number of new data and I think it should be acceptable after minor corrections.

We appreciate your positive observations about our work and are grateful for all your comments.

Comments on the Quality of English Language: The langage is correct. 

Round 2

Reviewer 1 Report

Comments and Suggestions for Authors

the authors have addressed the questions and it is ready to publish.